# Modeling the Production of Microalgal Biomass in Large Water Resource Recovery Facilities and Its Processing into Various Commodity Bioproducts

James Pierson [1], Gopi Raju Makkena [1], Sandeep Kumar [2], Vinod Kumar [3], Vivekanand Vivekanand [4], Hasan Husain [1], Muhammad Ayser [1] and Venkatesh Balan [1,*]

[1]   Department of Engineering Technology, Cullen College of Engineering, University of Houston, Sugarland, TX 77479, USA; gopirajumakkena1997@gmail.com (G.R.M.); hhusain3@cougarnet.uh.edu (H.H.); mayser@cougarnet.uh.edu (M.A.)
[2]   Department of Civil and Environmental Engineering, Old Dominion University, Norfolk, VA 23529, USA; skumar@odu.edu
[3]   School of Water, Energy and Environment, Cranfield University, Cranfield MK43 0AL, UK; vinod.kumar@cranfield.ac.uk
[4]   Centre for Energy and Environment, Malaviya National Institute of Technology, Jaipur 302017, India; vivekanand.cee@mnit.ac.in
*   Correspondence: vbalan@uh.edu

**Abstract:** Algae are capable of sequestering nutrients such as nitrates and phosphates from wastewater in the presence of sunlight and carbon dioxide ($CO_2$) to build up their body mass and help combat climate change. In the current study, we carried out different case studies to estimate the volume of algal biomass that could be produced annually using the rotating algal biofilm (RAB) method in three large-scale water resource recovery facilities (WRRFs) in Texas: Fort Worth, Dallas, and Houston. We calculated the total amount of lipids, carbohydrates, and proteins that could be fractionated from the algal biomass while using the hydrothermal flash hydrolysis process, followed by converting these biomolecules into commodity products via reported methods and yields. In the first case study, we estimated the amount of biogas and electricity produced in anaerobic digesters when the algal biomass and sludge generated in large-scale WRRFs are co-digested. Using this approach, electricity generation in a large-scale WRRF could be increased by 23% and $CO_2$ emissions could be further reduced when using biogas combustion exhaust gases as a carbon source for the RAB system. In the second case study, it was estimated that 988 MT mixed alcohol or 1144 MT non-isocyanate polyurethane could be produced annually from the protein fraction in the WRRF in Fort Worth, Texas. In the third case study, it was estimated that 702 MT bio-succinic acid or 520 MT bioethanol could be produced annually using the carbohydrate fraction. In the fourth case study, it was estimated that 1040 MT biodiesel or 528 MT biocrude could be produced annually using the lipid fraction. Producing renewable commodity fuels and chemicals using the algal biomass generated in a WRRF will help to displace fossil fuel-derived products, generate new jobs, and benefit the environment.

**Keywords:** microalgal biomass; $CO_2$ sequestration; wastewater treatment; rotating algal biofilm; commodity bioproducts

## 1. Introduction

Effectively treating wastewater is important to combat environmental pollution, eutrophication, and to facilitate the feasibility of recycling water [1,2]. Globally, it is estimated that nearly 360 billion $m^3$ of wastewater is generated annually [3]. Each water resource recovery facility (WRRF) uses different unit operations to treat the incoming wastewater from various sources [4–7], with the most common sources being domestic, agriculture, and processing industries. Depending on the source of the water or the design of the sewage system, the composition of the wastewater can vary greatly [8]. Common contaminants

found in wastewater include microorganisms, metals, organic or inorganic materials, and nutrients such as nitrogen, phosphorus, and ammonium. Table 1 lists the composition of wastewater originating from different sources and the EPA requirements for their discharge.

**Table 1.** The compositions of various wastewater sources and EPA's requirements for discharge.

| Composition | Municipal Wastewater | Industrial Wastewater | Agricultural Wastewater | EPA Requirements for Discharge |
|---|---|---|---|---|
| BOD (mg/L) | 200 | 1000–2000 | 3000–4000 | 50 |
| COD (mg/L) | 500 | 1000–1500 | 1000–5000 | 250 |
| TN (mg/L) | 40 | 50–100 | 200–400 | 50 |
| TP (mg/L) | 10 | 10 | 50–100 | 2 |
| TDS (mg/L) | 500 | 1000–10,000 | 500–5000 | 1500 |
| pH | 7 | 8–9 | 6–7.5 | 6–9 |

In 2019, the Texas Commission on Environmental Quality reported that Texas WRRFs treated approximately 5.4 billion gallons of wastewater per day. To treat such a high volume of wastewater, 5700 permitted WRRFs (including municipal and industrial wastewater, and other facilities) were established [9]. Currently, wastewater is collected from the source and transported to a local WRRF via sewage systems, where the water undergoes numerous processing steps to treat and recycle the water back into society or let it out into waterways. Most WRRFs are composed of primary, secondary, and tertiary treatment stages that use physical, chemical, and biological methods such as bacteria or their combination (Figure 1A). Nevertheless, all processes generate significant amounts of $CO_2$ in the environment. These treatment stages remove suspended solids contaminants from the water to reduce biological oxygen demand (BOD) and chemical oxygen demand (COD) before discharge [10]. An emerging alternative biological method is using algal technology. Unlike common tertiary treatments (e.g., flocculation, tertiary filters, or chlorine disinfection), algae sequester nitrates and phosphates from wastewater without producing carcinogenic byproducts [11,12]. The rotating algal biofilm (RAB) reactor system is exclusively developed to grow algae and can be used in multiple areas [13]. RAB-integrated WRRFs can be utilized to treat wastewater in the form of secondary or tertiary treatments via nutrient removal, industrial pretreatment, and side stream treatment such as anaerobic digestion effluents, as well as producing algal biomass, as shown in Figure 1B. The RAB system is composed of a rotating biofilm made up of a natural fabric like cotton or a polymer material like nylon, which rotates between an air phase and a wastewater liquid phase.

The RAB system is built in an enclosed room such as a greenhouse to treat wastewater to reduce the impact of external climatic conditions. When the fabric is in contact with wastewater, microalgae grow on the fabric and absorb the nutrients present in wastewater. As the fabric rotates into the air phase, the microalgae attached to the fabric can be harvested through a simple harvesting process of scraping them off. This is one of the benefits of RAB systems compared to other algae cultivation systems, such as raceway ponds or photobioreactors, which require energy-intensive centrifugation or filtration processes to separate algae [14]. Once harvested, the microalgae produced from RAB systems can be used as a feedstock, either using thermochemical or biochemical processes, for producing biofuels and renewable chemicals in a biorefinery. To create these biofuels and renewable chemicals, the generated algal biomass needs to be fractionated into proteins, lipids, and carbohydrates. Flash hydrolysis (FH) is the preferred hydrothermal treatment method for achieving this since the use of chemicals is not required to lyse the algal cells.

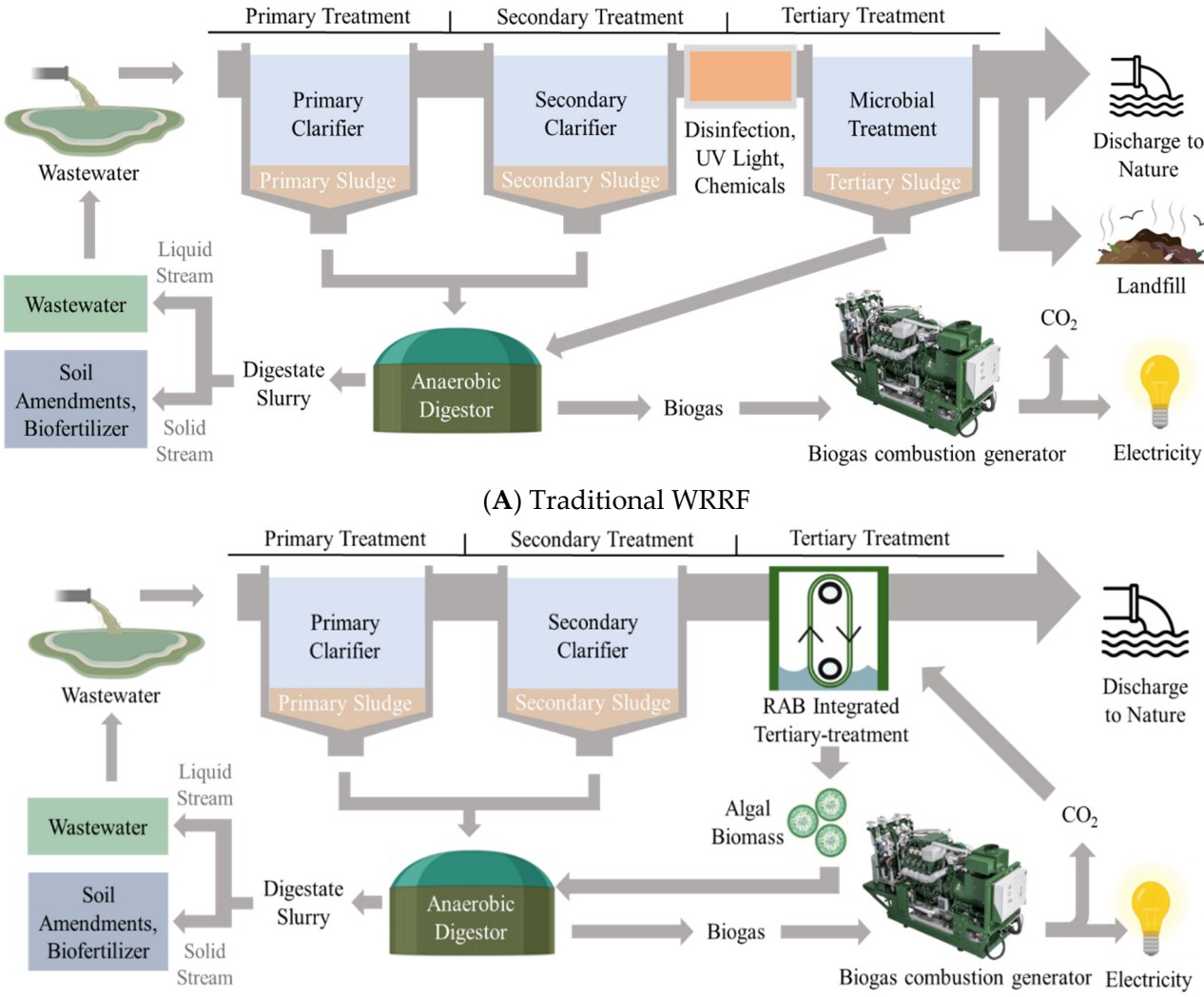

**Figure 1.** Traditional and new WRRF system. (**A**) Traditional WRRF process using microorganisms, and (**B**) RAB-integrated WRRF where algal biomass is used in the AD process to produce biogas and soil amendment.

In this study, the method of producing algal biomass from wastewater using the RAB system is described in detail. We have estimated the amount of algal biomass that could be potentially produced in three large WRRFs in Texas, such as Fort Worth, Dallas, and Houston. Four case studies have been performed under industrially relevant conditions using reported processes to evaluate the potential of manufacturing various commodity fuels and chemicals from algal fractions such as proteins (polyol or mixed alcohol), carbohydrates (bioethanol or bio-succinic acid), and lipids (biodiesel or biocrude). This study is the first of its kind and will lay the foundation for exploring existing WRRF infrastructure in the US and other parts of the world to produce commodity fuels and chemicals. This will help the transition from the current petroleum-based economy to a biobased economy that is sustainable and renewable and will benefit the environment and economy in many ways. We used a linear model to identify the potential revenue streams by producing various products while processing algal biomass produced in WRRFs. However, the limitation of the model is that it does not account for production cost, yield and efficiency.

## 2. Materials and Methods

### 2.1. Algal Biomass Production Calculation

The potential weekly biomass production rate was calculated for each of the three major WRRFs in Texas: Fort Worth, Dallas, and Houston. The amount of wastewater processed by each WRRF per day is 138.9 million gallons in Fort Worth, 123.7 million gallons in Dallas, and 96 million gallons in Houston [15]. It has been reported that about 0.18 g of *Chlorella vulgaris* microalgae can be produced per liter (L) of wastewater every 7 days using RAB technology [16]. The potential algal biomass productivity from using RAB systems was calculated using this reported algal biomass yield (Equation (1)). This algal biomass has been reported to be used as feedstock for producing biogas to satisfy the energy needs of the WRRF [17]. The biochemical methane potential (BMP) was measured in a mixture containing 63% wastewater sludge and 37% wet algae slurry. Considering that every cubic meter of biogas can produce 2 kW/h of electricity, energy production can be quantified.

$$\text{Weekly Algal Biomass Yield} = (\text{gal of watewater})\left(\frac{3.79\ \text{L}}{1\ \text{gal}}\right)\left(\frac{0.18\ \text{g}}{1\ \text{L}}\right) \tag{1}$$

### 2.2. Hydrothermal Flash Hydrolysis Followed by the Fractionation of Algal Biomass

Algal biomass can be fractionated into various macromolecules using hydrothermal FH and then processed into valuable biofuels and biochemicals. The potential weekly yields of carbohydrates, proteins, and lipids from FH processing of the algal biomass can be predicted based on the previously calculated weekly algal biomass production rate [16]. The typical microalgal composition found in nature is 15–25% carbohydrates, 25–35% lipids, and 35–45% proteins [18]. The weekly yields of each macromolecule can be calculated using Equation (2):

$$\text{Weekly Macromolecule Yield} = (\text{Weekly Algal Biomass Yield})(\text{Algal Macromolecule Composition}) \tag{2}$$

### 2.3. Conversion of Algal Biomass Fractions to Various Commodity Products

Using the yields of each macromolecule, the yields for producing various commodity products can be calculated using reported methods. Proteins derived from microalgae have a 60% and 50% conversion efficiency to mixed alcohols and polyurethane foam, respectively [19–21]. Fermentable sugars derived from microalgae have a 51% and 72% conversion efficiency to bioethanol and bio-succinic acid, respectively [22,23]. Lipid-rich solids derived from microalgae have a 68.9% and 35% conversion efficiency to biocrude and biodiesel, respectively [24,25]. All commodity product yields can be calculated using Equation (3):

$$\text{Weekly Product Yield} = (\text{Weekly Macromolecule Yield})(\text{Conversion Efficiency}) \tag{3}$$

## 3. Results and Discussions

### 3.1. Algal Biomass Processing in Texas Biorefineries

The three largest WRRFs in Texas are found in Fort Worth, Dallas, and Houston. The largest of which (Fort Worth) can treat approximately 140 million gallons of wastewater per day [16]. Hypothetically, if an RAB system were to be used in Texas, the algal biomass productivity in the three large WRRFs in Texas could be calculated; this is given in Figure 2. After algal biomass is scraped off the fabric from the RAB system, they can be used as feedstock along with primary and secondary wastewater sludge in the AD as shown in Figure 1. Details about the biogas productivity when using algal biomass are further discussed in case study 1.

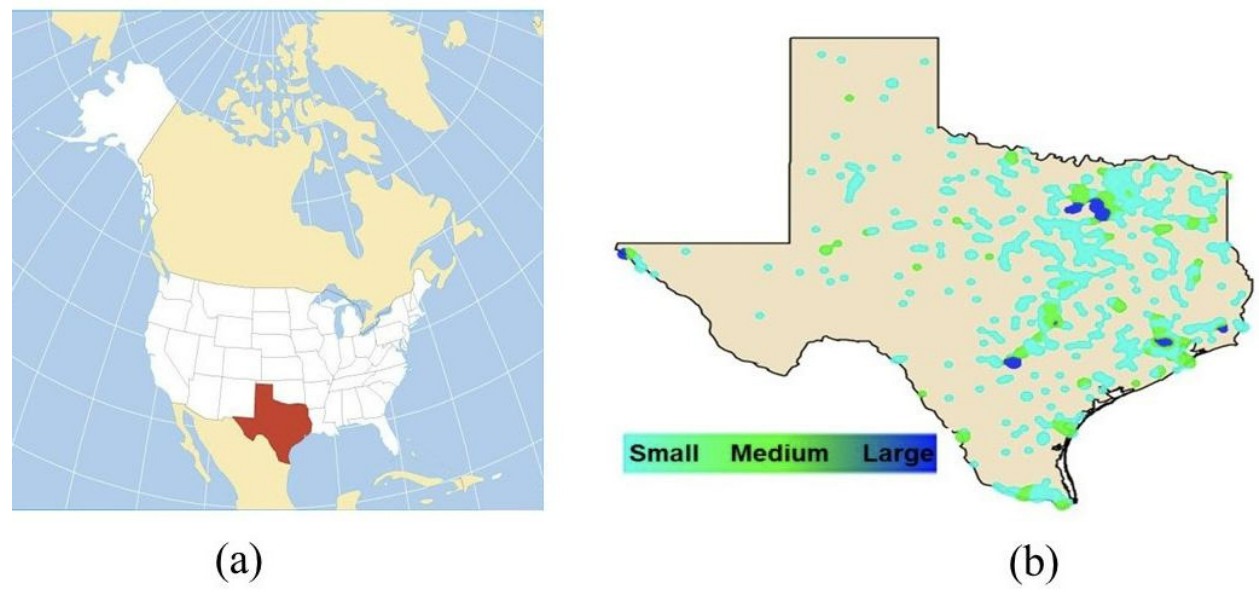

| Facility Location | Fort Worth | Dallas | Houston |
|---|---|---|---|
| Total Flow (MGD) | 138.9 | 123.7 | 96 |
| Weekly Biomass Production (0.18 g/L WW produced per week) | 94.76 MT | 84.3 MT | 65.4 MT |
| Monthly Biomass Production (30 days) | 406.1 MT | 361.2 MT | 280.2 MT |
| Annual Biomass Production | 4872.6 MT | 4334.7 MT | 3362.9 MT |

**Figure 2.** Location of Texas in north America and different wastewater treatment facilities in Texas. The capability of producing weekly, monthly, and annual algal biomass yields at three largest wastewater facilities located in Texas using RAB-Integrated WRRFs are given in the table above. MGD, million gallons of water per day; MT, metric ton (1000 kg). Here, (**a**) the map showing the United States and the state of Texas where the wastewater treatments plants are located, red region represents the state of Texas in the United States where the wastewater treatment plants are located and (**b**) location of small, medium and large wastewater treatment plants in Texas.

Alternatively, algal biomass can be collected and transported by a truck to a nearby biorefinery for further processing (Figure 3), which is discussed in case studies 2–4. The algal slurry arriving at a biorefinery will be subject to cell disruption to extract intracellular macromolecules. There are various methods of cell disruption techniques that microalgae can be subjected to, such as acid hydrolysis, enzymatic hydrolysis, or FH [26]. Of these mechanical cell disruption techniques, hydrothermal FH has several benefits as it is a chemical-free, continuous method that also increases lipid extractability. FH hydrolysate consists of a solid stream rich in lipids and a liquid stream rich in carbohydrates and proteins. The total amount of carbohydrates, proteins, and lipids that could potentially be extracted from algal biomass using the three major WRRFs in Texas is provided in Table 2. The FH hydrolysate is subject to centrifugation to separate the solid from the liquid stream. The liquid stream is subjected to acid or ammonium sulfate to precipitate the protein, which is separated by filtration. The separated proteins can be processed into non-isocyanate polyurethane (NIPU) foams or fermented to mixed alcohols using genetically modified *E. coli* [27]. The details of the protein processing steps are discussed in case study 2.

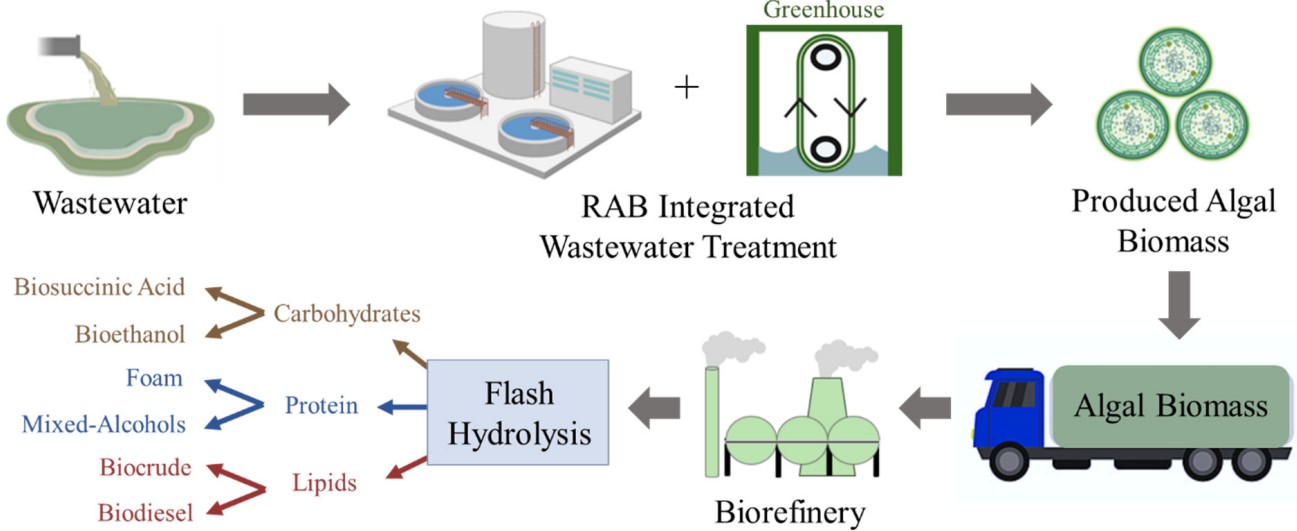

**Figure 3.** Logistics of producing and transporting algal biomass from RAB-integrated WRRF being transported to biorefinery by truck where they are subjected to flash hydrolysis and fractionated to carbohydrates, proteins, and lipids. The possibility of converting these fractions into commodity fuels, chemicals, and biomaterials is shown.

**Table 2.** Weekly algal carbohydrate, lipid, and protein yield from Texas wastewater facilities.

| Facility Location: | Fort Worth | Dallas | Houston |
|---|---|---|---|
| Total Flow (MGD) | 138.9 | 123.7 | 96 |
| Weekly Algal Biomass Yield | 94.76 MT | 84.3 MT | 65.4 MT |
| Weekly Carbohydrate Yield | 14.2–23.7 MT | 12.6–21 MT | 9.8–16.4 MT |
| Weekly Protein Yield | 33.2–42.6 MT | 29.5–37.9 MT | 22.9–29.4 MT |
| Weekly Lipid Yield | 23.7–33.2 MT | 21–29.5 MT | 16.4–22.9 MT |

The liquid stream after protein precipitation is further hydrolyzed using commercial enzymes to produce fermentable sugars, which could be fermented to bioethanol using yeast [22] or bio-succinic acid using native or genetically modified bacteria [23]. The details about processing steps and yield are further discussed in case study 3. The solid stream rich in lipids could be hydrothermally processed to lipids after extraction using nonpolar solvents, such as hexane, followed by transesterification to produce biodiesel [24] or biocrude [25], which is further discussed in case study 4. Separating the proteins from algal biomass reduces the amount of nitrogen incorporation in biocrude, which facilitates the downstream catalytic conversion process.

*3.2. Case Study 1: Algal Biomass to Biogas Conversion*

The algal biomass produced in the tertiary treatment step could be used as feedstock in the AD process to produce biogas. Conventionally, the sludge generated from the primary and secondary treatment step is used in the AD process to produce a biogas composed of methane (50–75%) and $CO_2$ (25–50%), in addition to smaller amounts of nitrogen-containing compounds, such as amines, amides, alkyl nitrates, alkyl nitrites, nitrosamines, nitroarenes, and peroxyacyl nitrates (2–8%), and trace amount of hydrogen sulfide ($H_2S$) [28,29]. After the AD process, the digested slurry undergoes solid/liquid separation, where the liquid stream (~60%) is recycled back to the WRRF, and the nutrient-rich solid stream (40%) comprising of nitrogen, phosphorus, and potassium can be dried and sold as a soil amendment [30,31]. Biogas is scrubbed using methods such as membranes, pressure swing adsorption, amine scrubbing, and water washing. It is then dried and combusted in a generator to produce electricity. Every cubic meter of biogas can produce 2 kW/h of electricity that will be used to satisfy the energy needs of a WRRF [17]. It is

important to note that the biogas yield and the composition of AD digestate can vary depending on the composition of the sludge and the processing conditions. Hypothetically, if a RAB system is used to treat the tertiary wastewater stream, the harvested algal biomass could be combined with the sludge produced in primary and secondary water treatment stages and used as feedstock for the AD process to produce biogas as shown in Figure 4. Most AD processes at WRRFs operate under mesophilic conditions around 37 °C, which is favorable for microalgae. Under these conditions, the co-digestion of microalgae with sewage sludge can increase biogas yield [32].

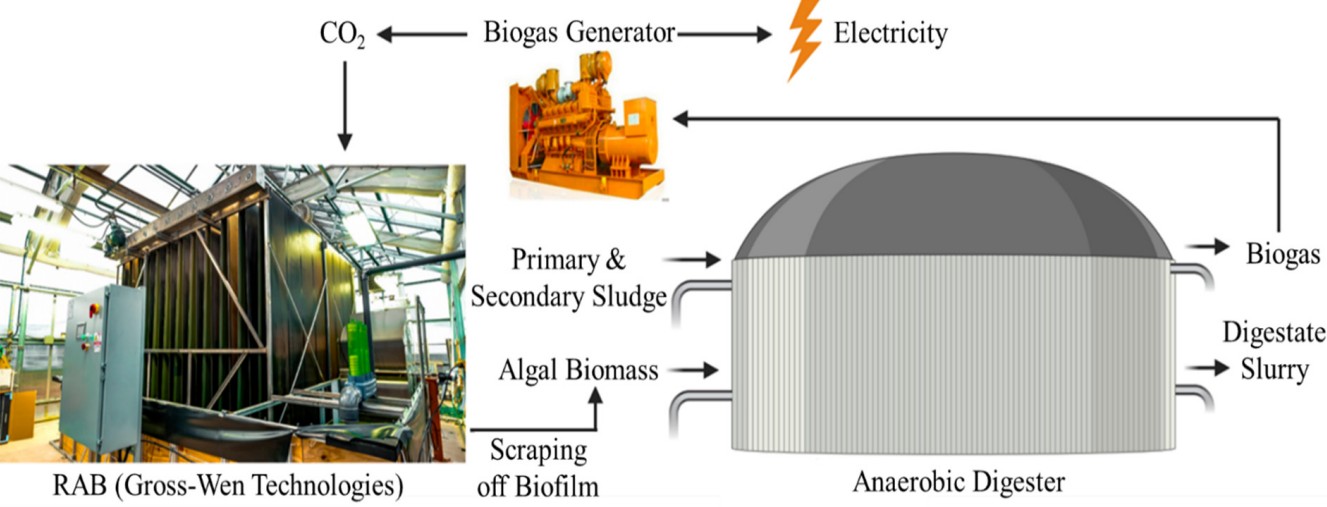

**Figure 4.** Close loop cycle of utilizing $CO_2$ from biogas combustion to reduce carbon emissions. This integrated process is expected to increase algal biomass yield and biogas yield.

AD is a biological process by which microorganisms break down biodegradable (organic) matter under anaerobic conditions and rely on the activity of diverse microbial communities to ultimately produce methane-rich biogas [33]. Microorganisms utilize metabolic processes to fully reduce carbon in organic material to methane gas, with $CO_2$ as a byproduct. Four microorganism groups contribute to this conversion, namely (i) hydrolyzers, (ii) acidogens, (iii) acetogens, and (iv) methanogens [33]. Hydrolyzers break down large organic compounds into smaller monomeric units through extracellular enzymes. There is a wide range of microorganisms capable of hydrolyzing organic compounds, such as *Pseudomonas* sp. or *Hartmanella* sp. The produced monomeric products are fermented by acidogens, yielding a mixture of long-chain fatty acids (LCFAs) and volatile fatty acids (VFAs) [34]. The produced fatty acids serve as intermediates and are utilized by acetogens to produce acetate, with $CO_2$ and $H_2$ as by-products through secondary fermentation. Acetogens are a diverse phylogenetic group of bacteria and have metabolic pathways which convert the produced fatty acid intermediates into acetate via fermentation. The fourth and final step in the AD process is methanogenesis, where methanogens utilize the produced acetate (or $CO_2$ and $H_2$ in small amounts) from acetogenesis to yield methane gas through fermentation [35]. Methanogenesis is often the rate-limiting step in AD. The BMP was measured in a mixture containing 63% wastewater sludge and 37% (*w/w* VS) wet algae slurry, which resulted in a 23% increase in methane yield compared to the wastewater sludge alone [32,36]. It should be noted that no increase in methane yields was reported when microalgae and wastewater sludge were co-digested under thermophilic conditions.

While producing electricity, the biogas combustion exhaust gas will be composed of $CO_2$ (13%) and water (13%), with nitrogen from air comprising the largest component (73%) at ambient conditions. However, when combusted using pure oxygen, the exhaust gas will comprise >85% $CO_2$ [37]. Injecting either of the two biogas exhaust gases into the RAB system will help to sequester the $CO_2$. This approach is expected to increase algal

biomass productivity [38,39]. Since the RAB system is capable of absorbing $CO_2$ during its air rotation, increasing the $CO_2$ concentrations inside the RAB greenhouse through a biogas generator exhaust will allow for the buildup of $CO_2$, which will be utilized by microalgae. The literature reveals a theoretical 200% algal biomass increase under $CO_2$ concentrations of 100,000 ppm compared to ambient air. Generating more algal biomass and using it as feedstock for the AD process will generate more biogas and a 23% higher electricity generation potential. The biogas productivity may be 50% higher when using lysed algal cells when compared to using whole algae cells during the AD process. Additionally, it has been stated that $CO_2$ favors the production of biodiesel from microalgal–bacterial granular sludge; therefore, sequestering the $CO_2$ will help to improve the fuel production of the microalgae [40]. This carbon recycling approach will help to reduce overall $CO_2$ emissions from the WRRFs and will help to combat climate change.

### 3.3. Case Study 2: Processing Algal Proteins into Mixed Alcohols and Polyurethane Foam

Microalgae are a rich source of protein, with compositions typically ranging from 20–50% in polycultures. Additionally, algal proteins have a wide range of industrial applications as they have been used in the production of high-protein aquaculture feeds, mixed-alcohol production, and the synthesis of hard foams. Since microalgal biomass is produced using wastewater, the processed algal proteins cannot be considered as a feed source as they may contain trace amounts of heavy metals. Instead, the algal protein will be further chemically processed into amino acids and fermented into mixed alcohols and polyols to produce non-isocyanate polyurethane (NIPU) foam.

As mentioned earlier, biomass cultivated from RAB systems in a large-scale WRRF can be transported to a biorefinery, where it will undergo FH, solid/liquid separation steps, and a protein precipitation step to isolate algal proteins. To produce mixed alcohols from algal proteins, they should be subject to dilute acid hydrolysis (2–4% sulfuric acid) or enzymatic digestion using proteases to produce individual amino acids, as shown in Figure 5 [41], and fermented using engineered *E. coli*. The engineered *E. coli* has upregulated amino acid catabolism pathways to convert protein fractions to higher C4 or C5 fusel alcohols, such as iso-butanol, 1-isopropanol, or n-butanol during fermentation [42]. Other byproducts that are produced during fermentation can be recovered after distillation [43]. The amino acids derived from the acid-hydrolyzed algal protein stream could be used as a source of polyols to produce NIPU foam. The FH hydrolysate is subjected to vacuum distillation to remove excess water and then treated with HCl, followed by neutralization using NaOH. As shown in Figure 5, the amino acids will react with excess ethylene diamine (EDA) to produce an amine-terminated intermediate, which is to be split into two streams producing cyclic carbonate and bifunctional salts, respectively [44]. Cyclic carbonate is produced by reacting the amine-terminated intermediate with ethylene carbonate, epichlorohydrin, and excess $CO_2$. The NIPU foam is produced when the bifunctional crosslinkers and cyclic carbonate are combined at room temperature, then raised to higher temperatures (120–160 °C) while mixing. The bifunctional salt releases $CO_2$ and water, allowing amine groups to react and form a polyurethane structure, which traps the released $CO_2$, forming a porous foam [45]. PU foams can also be efficiently produced from algal proteins using the isocyanate route. However, the non-isocyanate method is preferred as isocyanates are toxic chemicals that are extremely hazardous to humans and the environment.

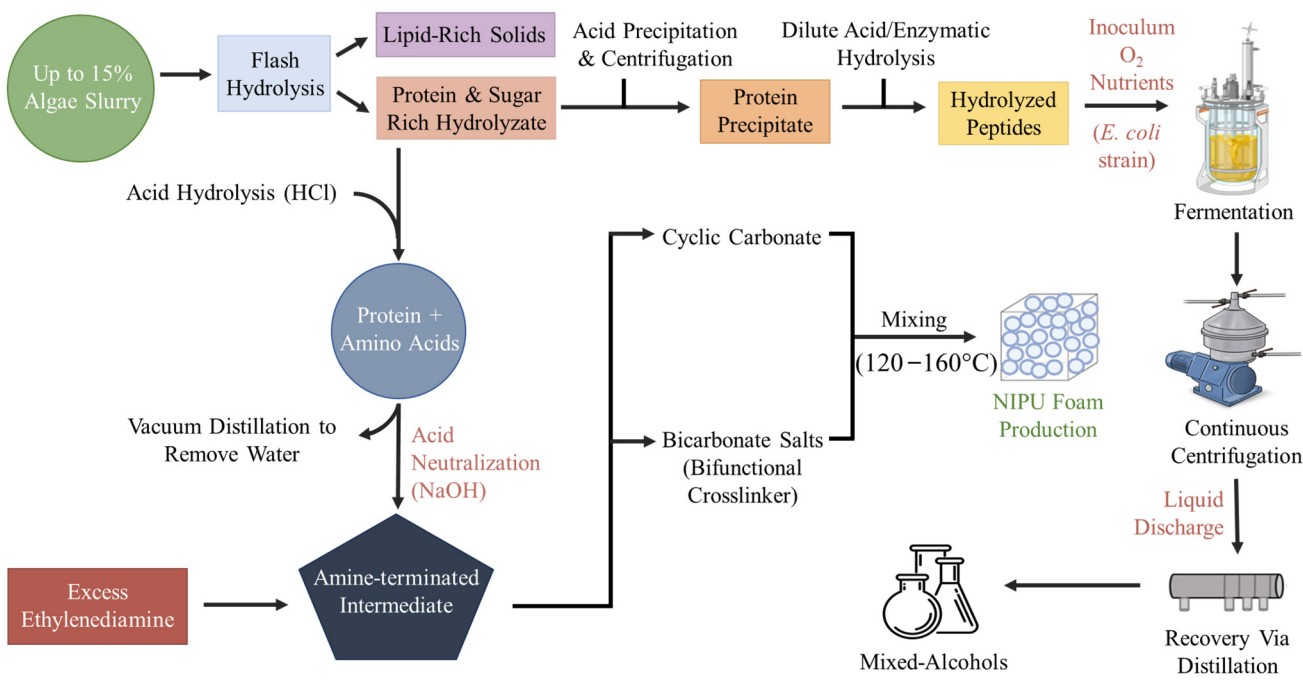

**Non-Isocyanate Polyurethane Foam from Amino Acids**

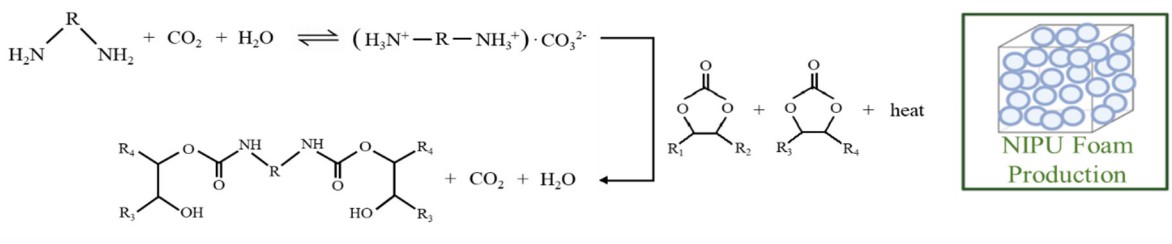

**Fermentation of Engineered E. coli for Mixed-Alcohol Production**

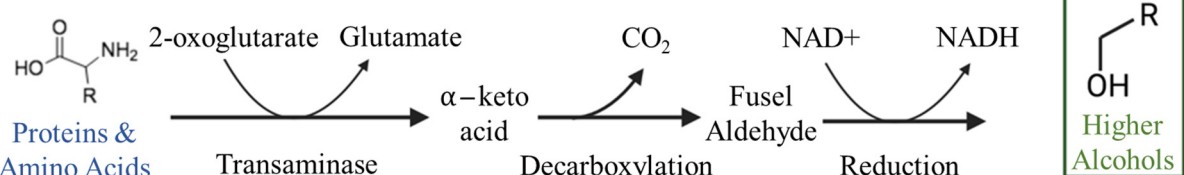

**Figure 5.** Conversion of algal proteins to produce NIPU foam and mixed alcohols via *E. coli* fermentation (**top**). The mechanism by which NIPU form and higher alcohol are produced are shown (**bottom**).

Currently, higher carbon alcohols are produced while processing crude oil in petroleum refineries. Producing bio-based mixed-alcohols, such as C4 and C5 (called fusel alcohols) using the biological route has several advantages when compared to petroleum-derived alcohols. Metabolically engineered *E. coli* is capable of deaminating proteins, allowing the production of C4 and C5 alcohols at approximately 60% yields at optimized fermentation conditions [46,47]. *E. coli* utilizes the Ehrlich pathway in protein fermentation, which converts branched-chain amino acids, aromatic amino acids, and sulfur-containing amino acids into fusel alcohols, such as isoamyl alcohol, iso-butanol, or propanol [48]. It is important to note that diamines can be produced via the decarboxylation of suitable amino acids and/or peptides, which are used in the production of both cyclic carbonate and bifunctional salts

in NIPU foam production. Hence, only 50% of microalgal proteins can be used in the conversion of amino acids to NIPU foam. The engineered *E. coli* can produce 4.04 g/L of alcohol from ~22 g/L of algal amino acids, rendering this a desired method for bio-based mixed-alcohol production [49].

Table 3 provides details about the amounts of mixed alcohols and NIPU foam that can be produced using algal protein processed from three large Texan WRRFs. Texas' largest WRRF, which is located in Fort Worth, can produce over 1100 tons of algal proteins annually that could be converted to 550 tons of NIPU foam annually based on a 50% conversation efficiency. With the global PU foam market projected to rise to over USD 41 billion by 2029, with hard foam and insulation applications, this clean and sustainable method should be further investigated for implementation [50].

**Table 3.** Projected microalgal-derived commodity products' annual yield from Fort Worth WRRF.

|  | Protein | | Carbohydrates | | Lipids | |
|---|---|---|---|---|---|---|
| Composition | 40 ± 5% | | 20 ± 5% | | 30 ± 5% | |
| Product | NIPU | Mixed-Alcohols | Bioethanol | Bio-succinic acid | Biocrude | Biodiesel |
| Conversion Efficiency | 50% | 60% | 51% | 72% | 68.9% | 35% |
| Maximum Product Yield | 988 MT | 1144 MT | 520 MT | 702 MT | 1040 MT | 528 MT |

*3.4. Case Study 3: Processing Algal Carbohydrates to Bioethanol and Bio-Succinic Acid*

Bioethanol and bio-succinic acid are two carbohydrate-derived biochemicals in high demand with a diverse range of applications. Primarily, ethanol is used as a fuel source, although it can also be utilized as a solvent in the chemical industry and sanitation in biomedical research fields [51]. In the fossil industry, ethanol is chemically derived from ethylene. The hydration of ethylene results in the formation of ethanol, which is achieved in industry using a reversible reaction between ethylene and water vapor [52]. However, with the rise of bio-based ethanol sources, the synthetic production of ethanol from ethylene is rapidly declining [53]. In 2021, bioethanol production in the United States totaled 15 billion gallons, which was primarily derived from corn starch [54]. In Brazil, bioethanol is produced using sucrose derived from sugarcane, where most of the ethanol is blended with up to 10% of petroleum products (E10 for regular vehicles), and this is up to 85% (E85 for Flex Fuel vehicles) in the US. In Brazil, 100% ethanol is used (E100 for power cars). Like corn, microbial biomass is rich in carbohydrates that can be converted to fermentable sugars. Carbohydrates found in algal biomass are predominantly long-chain polysaccharides, such as starch, cellulose, or hemicellulose. They must be hydrolyzed into free monomeric sugar molecules, such as glucose, xylose, and arabinose, using commercial enzymes. Typical enzymes used in hydrolyzing polysaccharides are CAZymes, beta-glucosidase, amylase, or pectinases. Hydrolyzed monomeric sugars are fermented to alcohol using native and genetically engineered yeast, such as *Saccharomyces cerevisiae* or *E. coli*, that can metabolize five carbo sugars in a fermenter, as shown in Figure 6 [48]. The produced ethanol present in the fermented slurry is subject to distillation, and dehydration using a molecular sieve to reach 99.5% purity, rectified with 5% methanol before being sold on the market.

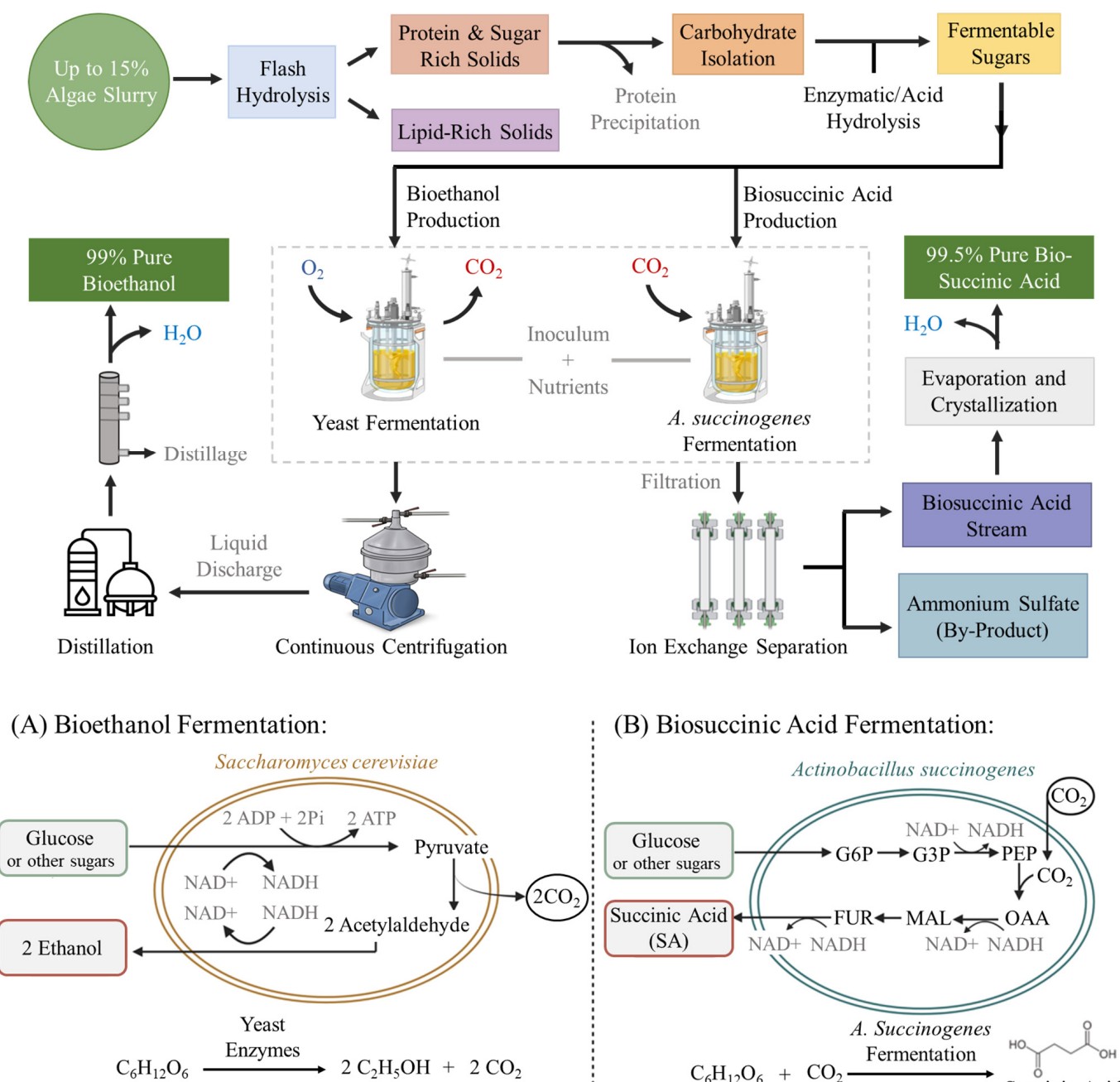

**Figure 6.** Processing steps to convert algal carbohydrates to fuels and chemicals (**top**). Pathways to produce bioethanol and succinic acid are shown (**bottom**). (**A**) Bioethanol fermentation using *Sacchromocyes cerevisiae*, and (**B**) bio-succinic acid fermentation using *Actinobacillus succinogenes* fermentation.

Similarly, bio-succinic acid is produced via bacterial fermentation. The U.S. Department of Energy (DOE) listed succinic acid as one of the twelve top sugar-derived building blocks in the market [55]. Succinic acid is a precursor to various industrial chemicals, such as adipic acid, maleic anhydride, or phthalic anhydride [55]. Currently, succinic acid is produced in the petrochemical industry and is a competitor to bio-succinic acid produced using corn starch. The petrochemical industry produces succinic acid through the hydrogenation of maleic anhydride to succinic anhydride, followed by the hydration of succinic anhydride to succinic acid [56]. However, as the economic feasibility of the fermentation process rises and climate concerns continue to impact the petrochemical industry, the bio-based

production of succinic acid continues to rise. The most common bacterial strain used in bio-succinic acid production is *Actinobacillus succinogenes* (found in the rumen digestive system). However, they can be produced using other genetically engineered bacterial strains, such as *E. coli* [57]. As seen in Figure 6, the fermented bacterial slurry is subject to filtration and an ion-exchange step following fermentation. This salt separation step yields a bio-succinic acid-rich liquid stream as well as an ammonium sulfate by-product, which can be purified and sold in the market as an additional revenue stream [58]. The remaining bio-succinic acid stream is subject to additional processing steps, such as crystallization and evaporation to remove excess water, resulting in 99.5% pure bio-succinic acid.

As mentioned earlier, various biological sources produce enough sugars capable of being used in the fermentation process to produce bio-based products, such as corn or microalgae. Table 4 compares the conversion efficiency of corn starch and sugars derived from algae in the production of bioethanol and bio-succinic acid. As seen in Table 4, corn starch to bioethanol conversion efficiency is as high as 86.5%. To produce fermentable sugar from corn, it should be subjected to milling and saccharification using alpha and glucoamylase commercial enzymes [58,59]. Conversely, approximately 51% of microalgae sugars are capable of being converted to bioethanol due to the presence of mixed sugars [22]. However, this number can vary depending on sugar composition, algae source, fermentation conditions, and product recovery techniques. Combining this information with theoretical RAB biomass productivity data from the three largest wastewater facilities in Texas enables the calculation of product yields for microalgae-derived bioethanol and bio-succinic acid, and the results are given in Table 4. Due to ranging algal biomass yields during RAB-integrated WRRF, varying carbohydrate compositions are present in algal polyculture, and ranging production conversion ratios, the standard deviations for carbohydrate yield, bioethanol production, and bio-succinic acid production are provided in Table 3. The standard deviation was calculated using varying biomass and product yields from the literature, including high, average, and low RAB biomass yields, carbohydrate compositions, and product conversion rates.

**Table 4.** Biochemical conversion of algal carbohydrates to produce SA and bioethanol.

| Feedstock | Product | Conversion Efficiency (%) | Yield (per MT Starting Material) | 100% Conversion Efficiency (MT) | 90% Conversion Efficiency (MT) | 80% Conversion Efficiency (MT) |
|---|---|---|---|---|---|---|
| Corn Starch | Bioethanol | 86.5 | 0.865 | 0.87 | 0.8 | 0.67 |
| Microalgae Sugars | Bioethanol | 51 | 0.51 | 0.51 | 0.45 | 0.41 |
| Corn Starch | Bio-succinic Acid | 74 | 0.74 | 0.74 | 0.67 | 0.60 |
| Microalgae Sugars | Bio-succinic Acid | 72 | 0.72 | 0.72 | 0.65 | 0.58 |

*3.5. Case Study 4: Converting Algal Solid Stream Rich in Lipids to Biocrude and Biodiesel*

Algal biomass is a rich source of lipids, with the composition depending on the strains and growth conditions. Lipids in microalgae can be classified into two categories: neutral lipids, such as triglycerides or cholesterol, and polar lipids, such as phospholipids or galactolipids [60]. The use of these neutral lipids has been explored in the literature as a source for many sustainable fuels, such as biodiesel and biocrude. Petroleum products are the sole source of about half of the entire country's $CO_2$ emissions [61]. Due to this, cleaner methods of producing biocrude and biodiesel by processing naturally derived lipids have been on the rise. The biodiesel production capacity in the U.S. reached 21 billion gallons per year in 2022, most of which is derived from soybean oil, used cooking oil, and tallow (animal fat) [62]. Nonetheless, microalgae have been proven to be a viable lipid source for producing biocrude and biodiesel.

As shown in Figure 7, the FH of microalgae yields a lipid-rich solid, termed the biofuel intermediate (BI), containing fatty acids and triglycerides capable of being transformed into industrial biocrude and biodiesel. For biodiesel production, algal lipids are extracted from the BI using solvents, such as hexane, then introduced to a biodiesel reactor where methanol is added as a substrate capable of reacting with triglycerides during the transesterification

process, yielding a mixture of fatty acid methyl esters (biodiesel) and the by-product glycerol [63,64]. Although other alcohol substrates can be used, methanol is commonly used in the industry due to its high availability and low cost. The transesterification process typically occurs in the presence of catalysts such as alkali, acid, or enzyme (lipase) to increase the rate of reaction [65]. The use of acid catalysts has proven beneficial in converting free fatty acids to methyl esters but are too slow for triglyceride conversion. For this reason, alkali catalysts, such as NaOH, are most used in industry as they are up to 4000× faster than acid catalysts [66].

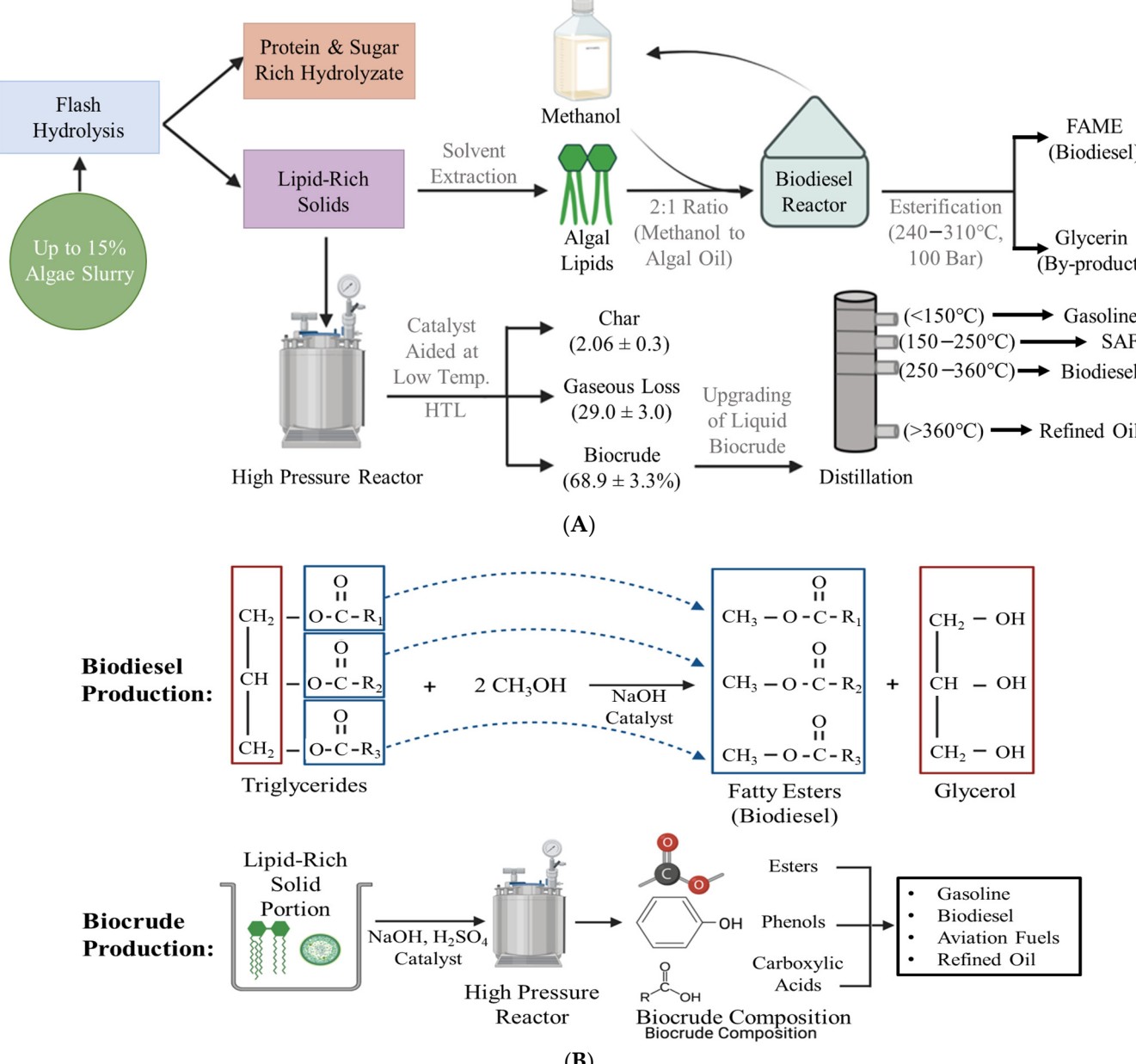

**Figure 7.** Method of converting lipid-rich algal biomass to produce biodiesel or biocrude. (**A**) Microalgae lipid conversion from the three largest Texas wastewater facilities, and (**B**) their respective product yields (biocrude and biodiesel).

However, free fatty acids from the BI may react with the added alkali catalyst during the transesterification process, forming a glycerol by-product [62]. Although the biodiesel yield from microalgae varies depending on the strain and transesterification conditions,

the transesterification of microalgae to produce biodiesel has been shown to have maximum yields of up to 35% when processed from *Spirulina* [67,68]. Table 5 compares the lipid-derived product yields of biodiesel and biocrude between microalgae and other traditional oil sources, such as used cooking oil. The biochemical conversion reaction steps of converting lipids to biocrude and biodiesel are given in Figure 7.

**Table 5.** Product conversion efficiency between algal lipids and cooking oil.

| Feedstock | Product | Conversion Efficiency (%) | Yield (per 1 MT Starting Material) | 100% Conversion Efficiency (MT) | 90% Conversion Efficiency (MT) | 80% Conversion Efficiency (MT) |
|---|---|---|---|---|---|---|
| Cooking Oil | Biocrude | 73 | 0.73 | 0.73 | 0.66 | 0.58 |
| Microalgae Lipids | Biocrude | 68.9–72.2 | 0.66–0.72 | 0.66–0.72 | 0.59–0.65 | 0.53–0.58 |
| Cooking Oil | Biodiesel | 36 | 0.36 | 0.36 | 0.32 | 0.29 |
| Microalgae Lipids | Biodiesel | 35 | 0.35 | 0.35 | 0.31 | 0.28 |

The BI produced from FH can be subject to hydrothermal liquefaction (HTL), which is a catalyst-aided high temperature and pressure reaction capable of depolymerizing lipid-rich biomass into crude oil [67]. Subjecting the microalgae BI to HTL yields $68.9 \pm 3.3\%$ biocrude, with additional solid residue and gaseous losses [62]. Interestingly, the biocrude yield following the HTL of the FH-produced BI was higher (68.9 wt.%) than the HTL of raw microalgae (43.3 wt.%), due to the increased lipid extractability of microalgae produced by FH [64]. Like petroleum-based crude oil, produced biocrude can undergo an upgrading process using hydrogen gas and can be distilled to isolate gasoline, aviation fuels, biodiesel, and refined oil based on different boiling points. Hypothetically, if microalgae were cultivated at large Texas wastewater facilities using RAB technology and the produced algal biomass were transferred to a biorefinery to undergo further processing, substantial amounts of biocrude and biodiesel would be able to be produced.

The market sizes and prices for different commodity chemicals and products that could be produced using algal biomass are given in Table 6 [69–71]. Some of the commodity chemicals, such as bio-succinic acid and polyurethane, have high market value, while other products, such as bioethanol, mixed alcohols, biodiesel, and biocrude, have low market value. It is important to note that while the scale-up process is not anticipated to follow a linear trajectory and each of these products have different production costs, it is evident that substantial revenue streams can be generated by incorporating the RAB system into WRRFs. Additionally, beyond the financial benefits, the production of fuels and chemicals using algal biomass as a feedstock has the potential to displace fossil fuel-derived products, thereby contributing positively to environmental sustainability, and it can also lead to the creation of new employment opportunities.

**Table 6.** Market size and product price of different fuels and chemicals that could be produced from algal biomass.

| | Source | Product | Market Size (MT) | Price (USD/MT) |
|---|---|---|---|---|
| Algae-based Biofuels | Lipids | Biodiesel | 25,000,000 | 1600 |
| | Carbohydrates | Bioethanol | 209,000,000 | 780 |
| | Lipids | Biocrude | 184,000,000 | 450 |
| Algae-based Bioproducts | Proteins | Polyurethane | 250,000 | 4980 |
| | Proteins | Mixed–Alcohols | 8,000,000 | 880 |
| | Carbohydrates | Succinic Acid | 2,300,000 | 3400 |

## 4. Conclusions

Producing bio-based products from RAB-integrated wastewater facilities brings environmental advantages as well as various revenue streams from produced products. From our modeling studies, we found that using microalgal biomass in an AD has the potential to increase biomethane yields by 23%. We also modeled the potential of transforming algal biomass produced in three WRRF in Texas to various high-value products. The WRRF in

Fort Worth, Texas, alone can produce 4872.6 MT algal biomass annually. This biomass could be processed to produce 528 MT of biodiesel or 1040 MT biocrude from algal lipids; 702 MT bio-succinic acid or 520 MT bioethanol from carbohydrates; and 1144 MT mixed-alcohols and 988 MT NIPU foam from proteins. As the influx of wastewater from numerous sources increases, the global demand for bio-based products will increase. The treatment of wastewater with algae and using algae biomass to manufacture commodity fuels and chemicals will help displace fossil fuel-derived products. This will be a viable option to help reach climate goals and benefit the economy. This linear model just predicts the benefits of using algal biomass in WRRFs and converting it to various fuels and chemicals. More detailed techno-economic assessments and lifecycle analyses of these processes will shed more light on the economic and environmental benefits.

**Author Contributions:** Conceptualization, V.B.; methodology, V.B., J.P. and G.R.M.; validation, S.K., V.K. and V.V.; formal analysis, V.B., J.P. and G.R.M.; investigation, J.P. and G.R.M.; resources, V.B.; data curation, J.P. and G.R.M.; writing—original draft preparation. V.B., J.P., V.K. and G.R.M.; writing—review and editing, S.K., M.A. and V.V.; visualization, H.H., J.P., G.R.M. and V.B.; supervision, V.B. All authors have read and agreed to the published version of the manuscript.

**Funding:** The research was made possible with funding from U.S. DOE, grant DE-FE0032203, and the UH Center for Carbon Management and Energy (CCME). This research is partially supported by the Texas Commission on Environmental Quality through an award to the Subsea Systems Institute. This project was paid for in part with federal funding from the Department of the Treasury through the State of Texas under the Resources and Ecosystems Sustainability, Tourist Opportunities, and Revived Economies of the Gulf Coast States Act of 2012 (RESTORE Act). The content, statements, findings, opinions, conclusions, and recommendations are those of the author(s) and do not necessarily reflect the views of the State of Texas or the Treasury.

**Institutional Review Board Statement:** Not applicable.

**Informed Consent Statement:** Not applicable.

**Data Availability Statement:** Not applicable.

**Acknowledgments:** We thank Martin Gross from Gross-wen technologies, Iowa, for helping us to obtain certain pieces of background information for this modeling study.

**Conflicts of Interest:** The authors declare no conflict of interest. The funders had no role in the design of the study; in the collection, analyses, or interpretation of data; in the writing of the manuscript; or in the decision to publish the results.

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
