# Peer review of "Modeling the Production of Microalgal Biomass in Large Water Resource Recovery Facilities and Its Processing into Various Commodity Bioproducts"

_fermentation, doi:10.3390/fermentation9100909_

Round 1

Reviewer 1 Report

I was very motivated to review this manuscript upon reading the title and the abstract - both of which teased the reader with the potential of a very revealing and contextualised study; however, upon reading the manuscript I was left somewhat disappointed with the product. Firstly, the manuscript suffers from an identify crisis that in many respects creates an overarching problem for this manuscript. The article is written (whether intentionally or not) as a weak hybrid between a review and a research article, except that the research content is flimsy. There article does not include the calculations/equations used to derive the values, where thereby makes validating the findings challenging. The authors talk about Case Studies - when I read a case study I expect to see the case study built on a foundation of solid data - this is not the case in this article, rather the authors present a series of 'scenarios' rather than case studies. There is a feeling to this article of a research proposal that has attempted to be repackaged as a research article. Regretably, for me, this manuscript fails to deliver and does not advance the literature.

Author Response

Response:

We thank the reviewer for spending his time to review our manuscript. We apologize that the manuscript did not meet his expectations, and we have taken your feedback to improve the paper.

Regarding the issue of identity, we would like to clarify that our manuscript is a modeling paper, heavily reliant on data reported in previous literature. We understand that the extensive literature citations may have created an impression of a hybrid between a review and a research paper. However, we have tried to integrate the two parts to make the paper as cohesive as possible.

In terms of calculations, they were explicitly described in the Materials and Methods section. However, to make it easier for readers to validate the findings, three equations have been added into the Materials and Methods section.

Finally, we want to assure you that the data in each case study is cited from reputable reports and peer-reviewed literature. Examples are given below:

  • “The amount of wastewater processed by each WRRF per day is 138.9 million gallons in Fort Worth, 123.7 million gallons in Dallas, and 96 million gallons in Houston [15].
  • It has been reported that about 0.18 g of Chlorella vulgaris microalgae can be produced per liter (L) of wastewater every 7 days using RAB technology [16].”
  • “The potential weekly yields of carbohydrates, proteins, and lipids from FH processing of the algal biomass can be predicted based on the previously calculated weekly algal biomass production rate [16]. The typical microalgal composition found in nature is 15-25% carbohydrates, 25-35% lipids, and 35-45% proteins [18].”
  • “Using the yields of each macromolecule, the yields for producing various commodity products can be calculated using reported methods. Proteins derived from microalgae have a 60% and 50% conversion efficiency to mixed alcohols and polyurethane foam, respectively [19-21]. Fermentable sugars derived from microalgae have a 51% and 72% conversion efficiency to bioethanol and bio-succinic acid, respectively [22,23]. Lipid-rich solids derived from microalgae have a 68.9% and 35% conversion efficiency to biocrude and biodiesel, respectively [24,25].”

Reviewer 2 Report

1. It is recommended that Figure 2 be replaced with a clearer picture.

2. The format of the chemical formula needs to pay attention to the subscript, such as CO2, etc.

3. There are many problems with the paragraph format in the manuscript, which needs to be checked and revised. In particular, Abstract and Conclusions are recommended to be written as a paragraph.

4. Results and Discussions suggest to do some more in-depth mechanism analysis.

5. Suggested references CO2 favors the lipid and biodiesel production of microbial-bacterial granular sludge[J]. Results in Engineering, 2023, 17: 100980.

Minor editing of English language required

Author Response

  1. It is recommended that Figure 2 be replaced with a clearer picture.

Response: We did put high quality figure and it looks good to us.

  1. The format of the chemical formula needs to pay attention to the subscript, such as CO2, etc.

Response: I think the reviewer is referring to mistakes in Figure 5. We have revised Figure 5.

  1. There are many problems with the paragraph format in the manuscript, which needs to be checked and revised. Abstract and Conclusions are recommended to be written as a paragraph.

Response: We checked all the paragraph format in the manuscript and made sure they are correct. We are not sure about the abstract and conclusion formatting. In our opinion they look good.

  1. Results and Discussions suggest doing some more in-depth mechanism analysis.

Response: This is a modeling manuscript and there is a certain limitation we have in providing in-depth mechanism analysis.

  1. Suggested references CO2 favors the lipid and biodiesel production of microbial-bacterial granular sludge[J]. Results in Engineering, 2023, 17: 100980.

Response: Thank you for providing this reference, which we have now included in the manuscript.

Reviewer 3 Report

I think that the article entitled “Modeling the production of microalgal biomass in large water resource recovery facilities and processing to various commodity bioproducts” presents results of interest to both the scientific community and the general population. However, some adjustments are necessary to improve the document.

Abstract

The abstract needs to be rewritten. I think this section should have more data to make it more attractive. Mainly the economic part of each case.

Keywords

I think the keywords are appropriate.

Introduction

I think that the agro-industrial sector also generates large quantities of effluents that must be considered both in terms of quantity and their physical-chemical characteristics. Therefore, I think they should be added to table 1. For example, review the following articles. Mainly because these wastewaters are sources of macro and micro nutrients that can be used as substrates for microalgae.

-          Agro-industrial wastewater in a circular economy: Characteristics, impacts and applications for bioenergy and biochemical.

-          Current developments and challenges of green technologies for the valorization of liquid, solid, and gaseous wastes from sugarcane ethanol production.

-          Biological wastewater treatment (anaerobic-aerobic) technologies for safe discharge of treated slaughterhouse and meat processing wastewater.

-          Co-digesting sugarcane vinasse and distilled glycerol to enhance bioenergy generation in biofuel-producing plants.

Furthermore, I think I should have had a more in-depth discussion of the physical and chemical characteristics of the effluents.

Algal biomass processing in texas biorefineries

I think that in this part it should be completely detailed how the biomass would be produced, processes, techniques, etc. in each case (1, 2, 3, 4).

Table 2. The calculations or estimations in this table must be made based only on one type or species of microalgae, what are they?

The authors of the work must be clear that the scale-up of processes is not linear, on the contrary, many variables must be considered. Therefore, I think that in the methodology, appropriate methodologies or appropriate models should be described for the projection of each of the study cases (1, 2, 3 and 4).

Currently, there are articles for the production of each of these metabolites. However, the problem is production costs. I think the exercise carried out is good, however it must be complemented with the technical and economic part in each case.

Conclusion

I think the conclusion should be rewritten after adjustments to the other sections.

References

I think the references used in the document are adequate.

Thank you so much

Author Response

Abstract

The abstract needs to be rewritten. I think this section should have more data to make it more attractive. Mainly the economic part of each case.

Response: Based on the reviewer’s feedback, we have made necessary changes to the abstract.

Introduction

I think that the agro-industrial sector also generates large quantities of effluents that must be considered both in terms of quantity and their physical-chemical characteristics. Therefore, I think they should be added to table 1. For example, review the following articles. Mainly because these wastewaters are sources of macro and micronutrients that can be used as substrates for microalgae.

  • Agro-industrial wastewater in a circular economy: Characteristics, impacts and applications for bioenergy and biochemical.
  • Current developments and challenges of green technologies for the valorization of liquid, solid, and gaseous wastes from sugarcane ethanol production.
  • Biological wastewater treatment (anaerobic-aerobic) technologies for safe discharge of treated slaughterhouse and meat processing wastewater.
  • Co-digesting sugarcane vinasse and distilled glycerol to enhance bioenergy generation in biofuel-producing plants.

Furthermore, I think I should have had a more in-depth discussion of the physical and chemical characteristics of the effluents.

Response: We tried to give three generic wastewater streams in Table 1. All the wastewater streams which the reviewer has outlines will fall under these three streams. Hence, we decided not to revise Table 1.

Algal biomass processing in Texas biorefineries

I think that in this part it should be completely detailed how the biomass would be produced, processes, techniques, etc. in each case (1, 2, 3, 4).

Response: Details were expanded upon in the following sections 3.2-3.5. I think that moving this information into section 3.1. would disrupt the natural flow of the paper.

Table 2. The calculations or estimations in this table must be made based only on one type or species of microalgae, what are they?

Response: The estimations presented in the table were based on the data presented in reference 16. In that study, the researchers used a microalgae polyculture in a raceway pond from the Algal Production Facility at Iowa State University (Boone, IA, USA) was used as seed for the RAB. While it was stated to be a stable indigenous microalgae consortium including prokaryotic and eukaryotic species, the main microalgae species present in the pond was Chlorella vulgaris. This has now been specified in section 2.1:

“It has been reported that about 0.18 g of Chlorella vulgaris microalgae can be produced per liter (L) of wastewater every 7 days using RAB technology [16].”

The authors of the work must be clear that the scale-up of processes is not linear, on the contrary, many variables must be considered. Therefore, I think that in the methodology, appropriate methodologies or appropriate models should be described for the projection of each of the study cases (1, 2, 3 and 4).

Response: A linear projection model was used in the case studies. It is true that the scale up process is linear, and this fact is now stated in the manuscript as a limitation of the study: “It is important to note that while the scale-up process is not anticipated to follow a linear trajectory and each of these products have production costs, it is evident that substantial revenue streams can be generated by incorporating the RAB system into WRRFs. Additionally, beyond financial benefits, the production of fuels and chemicals using algal biomass as a feedstock has the potential to displace fossil fuel-derived products, thereby contributing positively to environmental sustainability, and it can also lead to the creation of new employment opportunities.”

Currently, there are articles to produce each of these metabolites. However, the problem is production costs. I think the exercise carried out is good, however it must be complemented with the technical and economic part in each case.

Response: We acknowledged as a limitation of the study, but it is justified by its environmental and economic impact: “It is important to note that while the scale-up process is not anticipated to follow a linear trajectory and each of these products have production costs, it is evident that substantial revenue streams can be generated by incorporating the RAB system into WRRFs. Additionally, beyond financial benefits, the production of fuels and chemicals using algal biomass as a feedstock has the potential to displace fossil fuel-derived products, thereby contributing positively to environmental sustainability, and it can also lead to the creation of new employment opportunities.”

Conclusion

I think the conclusion should be rewritten after adjustments to the other sections.

Response: We have re-written the conclusion section.

Round 2

Reviewer 1 Report

This manuscript needs a section that explores the limitations of the modelling approaches used - there is no such thing as a perfect model and it is important that you inform readers of limitations of your approach. There was very little of this in evidence in the manuscript.

Author Response

 We added a sentence at the end of the introduction section about the limitation of the model ‘We used a linear model to derive the potential revenue streams by producing various products while processing algal biomass produced in WRRF. However, the limitation of the model is that it does not account for production cost, yield and efficiency’. We also added a sentence at the end of the conclusion section about future possibilities ‘This linear model just predicts the benefits of using algal biomass in WRRF and converting it to various fuels and chemicals.  More detailed techno economic assessment and life cycle analysis of these processes will shed more light on the economic and environmental benefits’. 

Reviewer 2 Report

accept

Author Response

Thank you for accepting the revised manuscript. 

Reviewer 3 Report

I think there were still some adjustments needed in the first revision document.

For example.

There was no discussion of the physical and chemical characteristics of the effluents.

The quantities of effluents by type were not observed.

In each case, economic data must be presented

Thank you very much

Author Response

Response: The scope of this modeling manuscript is to demonstrate the potential of producing algal biomass from water resource recovery facilities (WRRF) and converting it to various fuels and chemicals. More detailed experimental studies must be done in the future to accurately capture the effluents that are produced during each processing step which is outside the scope of this modeling manuscript.